# Quantification of amyloid fibril polymorphism by nano-morphometry reveals the individuality of filament assembly

Liam D. Aubrey[1,4], Ben J. F. Blakeman[1,4], Liisa Lutter [1], Christopher J. Serpell [2], Mick F. Tuite[1], Louise C. Serpell [3] & Wei-Feng Xue [1✉]

Amyloid fibrils are highly polymorphic structures formed by many different proteins. They provide biological function but also abnormally accumulate in numerous human diseases. The physicochemical principles of amyloid polymorphism are not understood due to lack of structural insights at the single-fibril level. To identify and classify different fibril polymorphs and to quantify the level of heterogeneity is essential to decipher the precise links between amyloid structures and their functional and disease associated properties such as toxicity, strains, propagation and spreading. Employing gentle, force-distance curve-based AFM, we produce detailed images, from which the 3D reconstruction of individual filaments in heterogeneous amyloid samples is achieved. Distinctive fibril polymorphs are then classified by hierarchical clustering, and sample heterogeneity is objectively quantified. These data demonstrate the polymorphic nature of fibril populations, provide important information regarding the energy landscape of amyloid self-assembly, and offer quantitative insights into the structural basis of polymorphism in amyloid populations.

[1] Kent Fungal Group, School of Biosciences, University of Kent, Canterbury CT2 7NJ, UK. [2] School of Physical Sciences, University of Kent, Canterbury CT2 7NH, UK. [3] Sussex Neuroscience, School of Life Sciences, University of Sussex, Falmer, Brighton BN1 9QG, UK. [4]These authors contributed equally: Liam D. Aubrey, Ben J. F. Blakeman. ✉email: W.F.Xue@kent.ac.uk

Amyloid fibrils are well known for their association with protein misfolding diseases, collectively known as Amyloidoses, as well as several neurodegenerative disorders. Each disease is characterised by a specific protein that accumulates in the peripheral tissues, in peripheral organs or in the brain. All amyloid fibrils have a core cross-β molecular architecture composed of β strands that stack perpendicular to the long fibril axis to form protofilaments up to several microns in length[1]. Multiple protofilaments can laterally associate to form twisted fibrils with a hydrophobic core[2,3]. Alzheimer's Aβ is one of the best known of the amyloid proteins, others include islet amyloid polypeptide in diabetes type 2[4], β-2 microglobulin in dialysis related amyloidosis[5–7], transthyretin in familial amyloidotic polyneuropathy[8,9] amyloid A (AA) in acute phase amyloidosis[10], tau in Alzheimer's and other tauopathies, and α-synuclein in Parkinson's disease[11]. There are also many known amyloid forming proteins that are not associated with disease but with biological function, including structural components of biofilms such as Curli expressed in *E. coli*[12,13], and amyloid fibrils involved in human skin pigmentation such as Pmel17[14,15].

Advances in high resolution structural determination techniques such as X-ray crystallography, solid-state nuclear magnetic resonance (ssNMR) and cryo-transmission electron microscopy (cryo-EM) have recently revolutionised our understanding of amyloid fibril structures[16–19]. For example, recent cryo-EM and ssNMR structures of Aβ[17,20,21] and tau[16,22,23] reveal parallel, in register organisation of the proteins within the cross-β core. These advances have provided molecular information on the 3D fold of the monomeric subunits within the core of distinct fibril types in addition to information about the intermolecular interactions between amino acid residues[24]. The impact of mutations on the core fold, for example in familial forms of Alzheimer's disease, can therefore be accessed[20]. Thus, on an atomic scale, these techniques have advanced our understanding of the core structures that make up individual amyloid fibril types.

On a mesoscopic (micrometre to nanometre) scale, amyloid fibrils display a high degree of polymorphism and amyloid populations are often highly heterogeneous. This has been demonstrated in disease related amyloid fibrils in both in vitro and ex vivo samples, as well as in samples of functional amyloid[25–27]. For example in a study involving Aβ, ssNMR data was compared for fibril structures taken from various patients which showed that the structures of the fibrils observed were highly varied from patient to patient in both Aβ$_{1–40}$ and Aβ$_{1-42}$[28]. Structural characterisation has revealed that many amyloidogenic proteins can form polymorphic structures, revealing that, for example, AA can fold into different conformations dependent on the species, as well as the individual patients[29], while tau has been shown to form different polymorphs in different tauopathies[16,22,30].

Structural polymorphism in amyloid fibrils, can be dependent on the number and orientation of the protofilaments that are arranged within a fibril[31], as well as the conformation of the monomeric subunits that make up the core of the fibril structure which is in turn dependent on the primary sequence[32]. Fibril structural polymorphism can be identified by the twists along the longitudinal axis of the fibrils producing varied periodic crossover distances, and by the shape and size of the cross-sectional area of the fibril[33]. Polymorphism is of interest due to the possibility that different fibril polymorphs may have different physical properties, such as the rate of fragmentation[34,35] or the ability to act as a catalytic surface for secondary nucleation[35,36]. Different physical properties could in turn result in different biological activities, such as the potential to propagate in a prion-like manner[37], the ability to associate with cytotoxic active species[38] or the impermeability of a biofilm matrix[39]. Another example of an amyloid structure function relationship is demonstrated by the difference in the structures of murine and human AA which results in a lower efficiency for the induction of amyloidosis between species[29]. Using electron microscopy (EM) and atomic force microscopy (AFM), it has been possible to observe and differentiate between polymorphs in amyloid fibril samples[27]. However, since fibril polymorphism has generally only been assessed in a qualitative manner with respect to the overall heterogeneity of amyloid fibril data sets[27], the structural, physicochemical and mechanistic origin of polymorphism, the extent to which it exists and its effect on biological activity is not known.

In order to identify, classify and quantify amyloid fibril polymorphism, as well as enumerate the heterogeneity of amyloid samples, the structure of individual fibrils in a sample population must first be resolved at sufficiently high-resolution so that individual twist patterns, twist handedness and cross-sectional profiles are distinguishable at a single-fibril level without cross-filament averaging. Here, using force-distance curve-based AFM imaging mode at low contact forces, we are able to image, in sufficiently high detail, individual fibrils made from three different peptides with sequences HYFNIF, VIYKI, and RVFNIM. These peptides are short amyloidogenic sequences originating within larger proteins (Human Bloom syndrome protein, Drosophila Chorion protein and Human elF-2, respectively[40]) identified via the Waltz algorithm[41]. Potential models of the amyloid protofilament core for each of the Waltz peptide assemblies have been presented previously (Fig. 1), which are validated using X-Ray Fibre Diffraction (XRFD) data, but these assembly reactions have also been previously observed to produce a large range of fibril polymorphs upon assembly[40], either through varied protofilament packing, changes in the core, or both. Thus, these short peptides are specifically selected as excellent model systems for investigating the structural basis of polymorphism within amyloid populations because of their shared tendency to form polymorphic amyloid samples despite having different primary sequences. Using the AFM image data sets collected, each individual fibril observed in AFM images are reconstructed as a distinct 3D model, and the structural parameters of individual fibrils in the sample populations are measured and compared. Finally, we employ an agglomerative hierarchical clustering method to classify fibril polymorphs by measuring the structural differences observed between individual fibrils. The results show that variation in the degree of structural polymorphism and the heterogeneity of amyloid fibril samples are highly sequence specific. More importantly, the data and analysis here demonstrate that that each individual amyloid fibril has characteristics that are different to those of the population average.

## Results

**Gentle AFM imaging identifies distinct fibril polymorphs to high detail.** Considerable heterogeneity has been displayed in several amyloid fibril samples previously[26,27], including in samples made from Waltz peptide amyloid assembly reactions[40]. To achieve the high level of detail required for quantitative structural analysis for individual amyloid fibrils without cross-particle averaging, we imaged the three Waltz peptide assembly samples, HYFNIF, RVFNIM and VIYKI, using a force-distance curve-based AFM imaging method. AFM is a high signal-to-noise method that has been used previously to image amyloid fibrils at high resolution[42–45]. Here, force-distance curve-based imaging (Peak-force tapping, ScanAsyst mode, Bruker) was employed rather than traditional tapping mode imaging so that the maximum force applied to the sample was kept consistent and minimal ensuring that the specimens were not deformed and that the surface details were tracked faithfully. Typical images of the

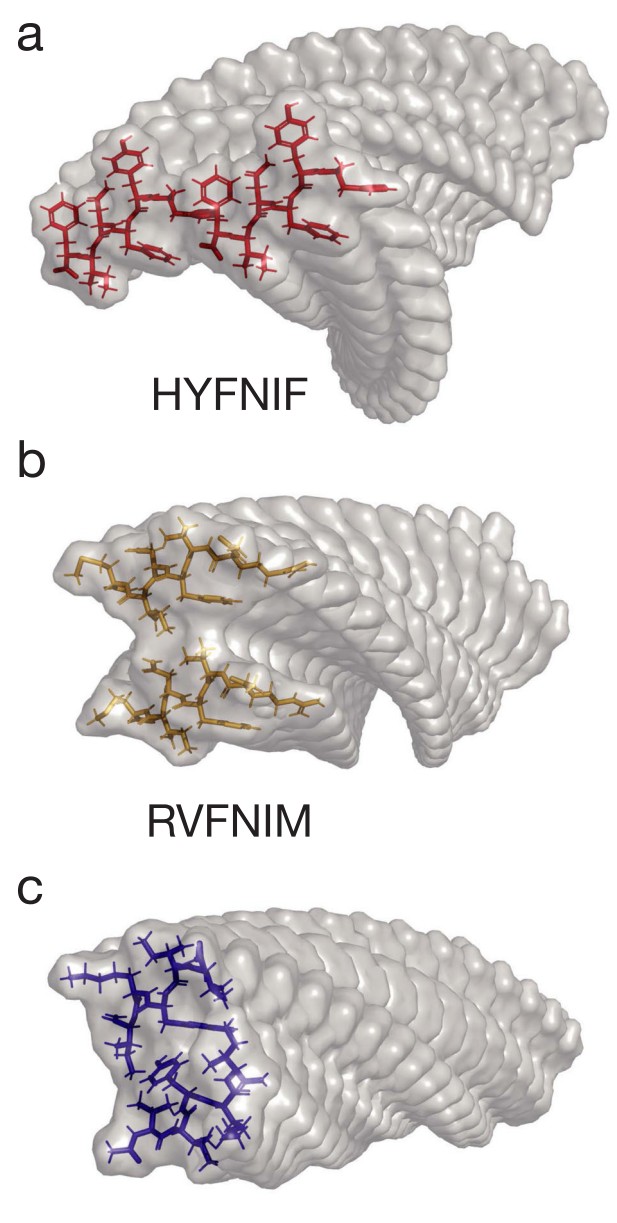

**Fig. 1 Amyloid core models of protofilaments formed from the three Waltz peptides.** Predicted models of amyloid protofilaments made from each of the 3 Waltz peptides, HYFNIF (**a**), RVFNIM (**b**) and VIYKI (**c**). These models show possible core packing in the protofilaments, and were generated and validated by comparing simulated and experimental XRFD data as detailed in Morris et al.[40].

three Waltz peptide assembly samples collected using force-curve based imaging are shown in Fig. 2. Closer inspection of these images shows that fine details such as the length and handedness of repeating patterns in the fibrils can be readily observed without any further processing of the raw image data. Qualitatively, it is clear that there is a high degree of heterogeneity in these samples since many fibrils of different heights and twist patterns, including some with different handedness can be observed from within the same image.

Around 90 well-separated fibrils for each of the Waltz peptide assemblies were chosen from the images. Their contours along the filament centres were individually traced and digitally

straightened. Figure 3 shows eight typical 500 nm long fibril segment examples from each of the three data sets (almost all traced fibrils were longer than 500 nm, only a portion of the traces are shown in Fig. 3). All three peptide precursors result in a set of visibly distinguishable, unique fibrils. All of the traced fibrils are displayed in Supplementary Fig. 1.

**3D reconstruction of individual amyloid fibrils**. An informative method for visualising and comparing polymorphic filament structures in high detail is to reconstruct 3D models of the filaments from the AFM image data. AFM has been a key imaging method that has enabled analysis of amyloid polymorphism[46,47]. Here, because AFM is a high signal-to-noise imaging method, we were able to extract structural coordinates from the AFM image data and reconstruct 3D structural models directly from these coordinates with sufficient detail without averaging across multiple filaments, thus producing individual models for each of the individual fibrils traced in our images. 3D reconstruction also allows for visualising and comparing filament structures without the influence of the varying degree of tip-sample convolution on the AFM images due to varying cantilever tip dimensions across different images (evident in the varied apparent width of the fibrils within the images in Fig. 3). Figure 4 shows all of the 266 3D models made, with examples of the 3D fibril models from the three data sets matching the images in Fig. 3 shown in Supplementary Fig. 2. In addition, the 3D models reveal the shape and size of the cross-section of each of the fibrils and how the cross-section rotates along the length of the fibril (Supplementary Fig. 3). In this respect, the VIYKI data set stands out as all of the fibril models show near-circular cross-sections. This is particularly striking when compared to the HYFNIF and RVFNIM data sets in which there are numerous fibril types with ellipsoidal cross-sections, as well as a few models which display asymmetric features. In summary, all three datasets display fibrils with varying degree of structural individualities, and a range of polymorph classes that can be distinguished qualitatively.

**Quantification of heterogeneity arising from fibril polymorphism**. Identifying the fibril polymorphs at the level of individual fibrils led to the qualitative observation that all three samples display a high degree of heterogeneity. In order to quantify, enumerate and compare the overall heterogeneity of the samples, we performed nano-morphometric measurements on individual filaments in our datasets. For each of the ~90 fibrils from each of the three datasets, six different structural parameters were measured (Supplementary Data, Fig. 5): maximum height ($h_{max}$), minimum height ($h_{min}$), average height ($h_{mean}$), handedness, periodic frequency, and average cross-sectional area ($csa$). The periodic frequency refers to the frequency of the most common repeating pattern observed in the height profile of a straightened filament identified by Fast Fourier Transform (FFT) of the height profile, corresponding to the twist pattern of the filaments. The frequency of the periodicity per nm was used rather than the periodicity as this would allow any twisted "ribbon" like fibrils with very long twist periodicity to be analysed and visualised in the same way as fibrils with a very short periodicity. The handedness of the twist was determined separately by manual inspection of the fibril images and 2D Fourier transform (2D power spectral density map) of the fibril images. The periodic frequency of fibrils with left-handed twists were assigned negative frequency values while the periodic frequency of right-handed twisted fibrils retained positive values to give directional periodic frequency ($dpf$) values that enabled the visualisation of the distribution between left and right handedness in each data set. The standard deviation (SD) of the 5-dimensional ($h_{mean}$, $h_{max}$, $h_{min}$,

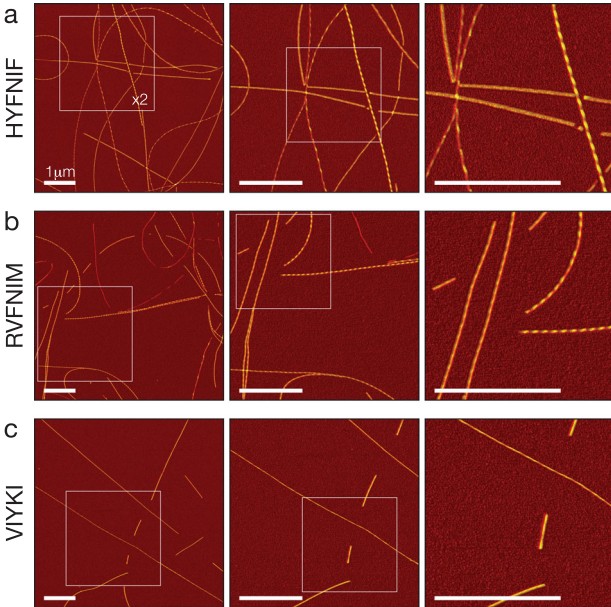

**Fig. 2 High resolution AFM imaging of Waltz peptide assemblies.** Fibrils generated by each peptide were deposited onto freshly cleaved mica and imaged using peak-force tapping mode AFM. Each row shows representative image data from each assembly reaction, HYFNIF (**a**), RVFNIM (**b**) and VIYKI (**c**). The original 6 μm by 6 μm images are shown in the left column and each subsequent column shows a 2-fold increase in magnification indicated by the white boxes. Different structural polymorphs are readily visible at high magnification including different twist patterns and height profiles. The colour scale represents the height range from 0 to 12.5 nm and the scale bar represents 1 μm in all images.

*dpf* and *csa*) Euclidian distance to the global mean for each dataset was subsequently calculated as a quantitative measure of sample heterogeneity (Eq. (2) in "Materials and Methods" section). For the three peptide sequences studied here, VIYKI peptide assembly showed the most structural heterogeneity with a SD of 2.57, RVFNIM had a SD of 1.98 and HYFNIF showed the least structural heterogeneity with a SD of 1.40.

In order to display the whole fibril population of morphometric measurements, the average height was plotted against the directional periodic frequency as a contour map which is shown in Fig. 6. The contours represent the density of the data in specific regions (seen as two-dimensional projections in Fig. 6). The information gained from the contour map is analogous to a plot of an energy landscape for the assembly reactions where deeper energy wells represent more likely structures since the probability that a system will be in a certain assembly state is a function of that state's free energy at a given temperature as given by the Boltzmann distribution. As shown in Fig. 6, both HYFNIF and VIYKI peptide assembly reactions appear to be dominated by the formation of left-handed twisted fibrils. Interestingly, in contrast, RVFNIM peptide assembly displays no overall dominant preference for twist handedness. There is, therefore, a sequence specific energetic favourability for forming predominantly left-handed or right-handed fibrils. In contrast to RVFNIM and VIYKI, HYFNIF fibril polymorphs were mostly populated in one narrow region of the contour map with a few outliers, suggesting that there was a deeper preference for one class of assembly compared to the other two sequences. Fibrils made from the VIYKI peptide however, covered a large range of polymorphs with different average heights and periodic frequencies, populating a large area in the contour diagram in Fig. 6, suggesting that there were multiple classes of structure that were almost equally

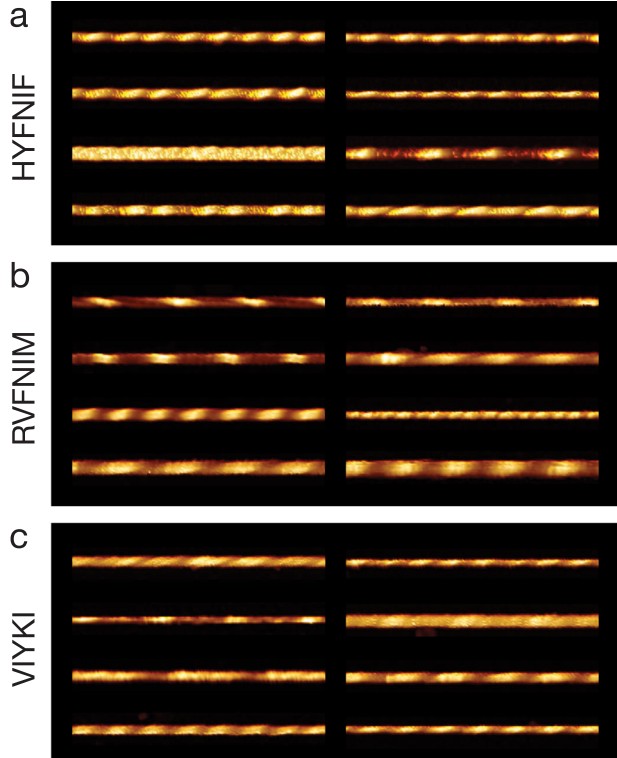

**Fig. 3 Comparison of distinct polymorphs from straightened fibril image data.** Around 90 fibrils from each data set were traced and 8 examples from each data set, HYFNIF (**a**), RVFNIM (**b**) and VIYKI (**c**), are displayed. The traced fibrils displayed here were straightened and cropped to 500 nm segments, and no further processing occurred. Qualitatively, each fibril can be distinguished from all of the other fibrils in each data set. Different twist patterns and average heights were readily visible. All of the fibril images analysed (around 90 fibrils for each data set) are displayed in the Supplementary Fig. 1.

likely to occur, and therefore similar in terms of the free energy associated with protofilament assembly. RVFNIM fibrils displayed an intermediate level of heterogeneity, with some preference for specific classes of both left-hand and right-hand twisted polymorphs. On a contour plot of the minimum height vs. the maximum height, shown in the Supplementary Fig. 4, VIYKI fibrils showed a tendency to have more similar values for minimum and maximum heights suggesting a preference for cross-sections with a particularly rounded shape whereas HYFNIF and RVFNIM assembled into fibrils that may have a larger maximum height than minimum height. This was also observed when the average cross-sectional areas of the models were plotted against the average height of the fibrils where the cross-sectional areas of VIYKI fibrils are distributed close to the line expected for circular cross-sections (cross-sectional area proportional to width squared). These observations support the qualitative assessment of the models in which the VIYKI fibrils overall appear to be more cylindrical. In summary, the three different assembly reactions, despite all resulting in amyloid fibrils with cross-beta core, display different assembly free energy landscapes, which are influenced by their side chain composition. This results in sequence specific preferences in the polymorphs that they form (e.g., in terms of width, twist and handedness) is due to variations in the packing of the side-chains within the protofilaments, the packing of the protofilaments in the fibrils or a combination of both, and results in differences in the heterogeneity of the fibril populations.

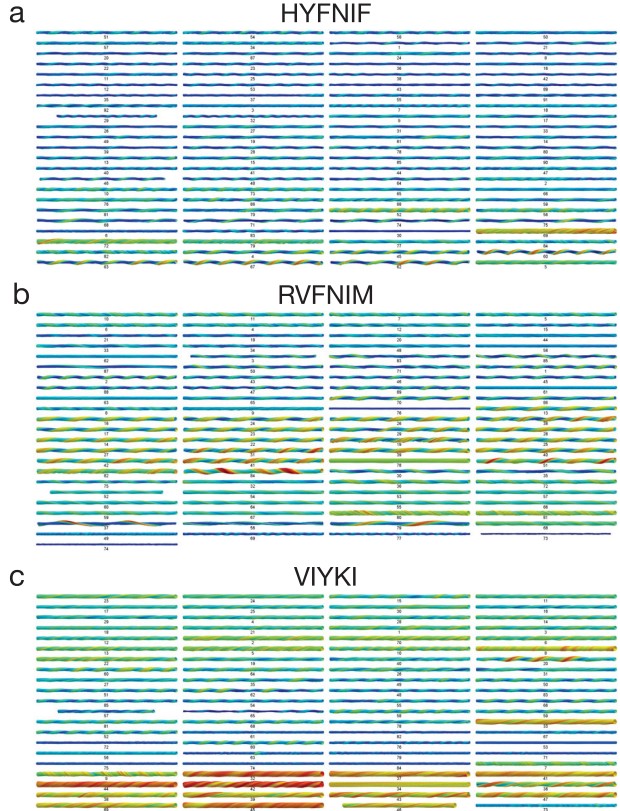

**Fig. 4 3D models of Waltz peptide assemblies.** Reconstructed fibril 3D models from each assembly reaction, HYFNIF (**a**), RVFNIM (**b**), and VIYKI (**c**) are displayed. Each fibril in the three data sets was reconstructed as a 3D models using information and 3D coordinates extracted directly from the AFM data. The average cross-sectional area and the helical symmetry was determined from the generation of each 3D model. All of the models are shown with identical scale and the colour represent the local radius to the screw axis for visualisation. All fibril models displayed here are cropped to 500 nm segments for visualisation if the contour length is longer than 500 nm. The models are shown with their individual index numbers used throughout (see Supplementary Data), are arranged by similarity (see Fig. 7 and Supplementary Fig. 5), and are placed in the same order as Supplementary Fig. 1. A selection of the models matching the images shown in Fig. 3 are displayed in the Supplementary Fig. 2.

### Amyloid fibril polymorphs classified by hierarchical clustering.

Classifying fibril polymorphs within a sample population can provide visualisation and organisation of the single-fibril level structural data that allows for further analysis of the activities and behaviours of types of structural polymorphs. Here, natural divisions in the data set of amyloid fibrils can be found by identifying which fibrils have shared or similar features and which fibrils do not, as determined by the nano-morphometric measurements. Subsequently, agglomerative or 'bottom-up' hierarchal clustering was performed to resolve the natural divisions within each dataset and therefore objectively classify the fibrils. The standardised Euclidean distance between each possible pair of fibrils in 5-dimensions (average height, maximum height, minimum height, directional periodic frequency and cross-sectional area), was calculated and linked together iteratively to generate a linked tree representing the structural and morphological relationships between individual fibrils. The morphometric data can then be represented in a dendrogram with the clusters on the x-axis and the 5-dimensional standardised Euclidian distance from one cluster to the next on the y-axis. In the

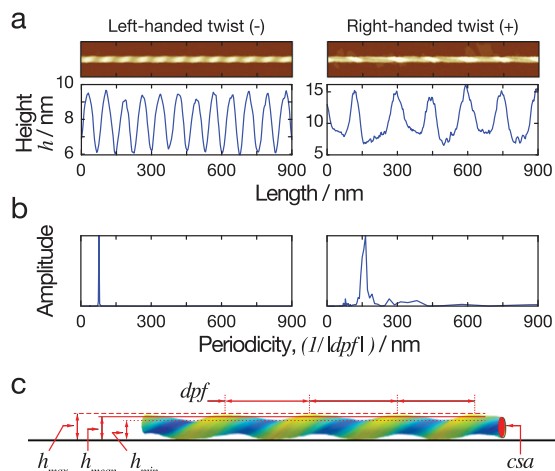

**Fig. 5 Nano-morphometry on individual fibrils results in quantitative structural parameters. a** Two example images of straightened fibrils, including a left-hand twisted fibril and a right-hand twisted fibril, are shown together with their corresponding height profiles across the centre line of each straightened fibril. The minimum, maximum and average heights can be determined directly from the height profiles. **b** FFT of the height profiles are shown, with peaks represent the periodicity describing the repeating units in the height profile. The average length covered by the repeating unit, representing the periodicity of the fibrils, was extracted from this analysis. Because the two example fibrils have different twist handedness, the directional periodic frequency (*dpf*) was assigned as a negative value for left-hand and a positive value for right-hand twisted fibrils. **c** A schematic diagram illustrating the quantitative structural parameters obtained from each individual fibril. The solid black line at the bottom representing the mica surface during AFM imaging. The parameters measured include the average height ($h_{mean}$), the minimum height ($h_{min}$), the maximum height ($h_{max}$), the directional periodic frequency (*dpf*), and the cross-sectional area (*csa*).

Supplementary Fig. 5, full dendrograms displaying the entire trees of the three datasets are presented. In order to separate the fibrils into 'classes', the data must be analysed at appropriate distance cut-offs. The distance cut-off determines the maximum distance within any clusters (or class). Therefore, the number of clusters at a given distance cut-off is indicative of the heterogeneity of the data set. A greater number of clusters suggest that the data is more spread out and therefore more heterogeneous. The dendrograms shown in Fig. 7a, which have been collapsed based on a distance cut-off of 1 (standardised Euclidean distance), show how the datasets were subsequently classified in Fig. 7b, c.

In total, at a distance cut-off of 1 in the standardised 5-dimensional Euclidean space, 13 separate clusters were generated by HYFNIF polymer assembly, 22 clusters were generated by RVFNIM polymer assembly and 19 clusters were generated by VIYKI polymer assembly. These clusters can be seen as classes of fibrils, where member fibrils of the same class show similar structure in terms of their morphometric appearances. Further analysis of the data at different distance cut-offs, shown in Supplementary Fig. 6, corroborate that HYFNIF fibrils display considerably less heterogeneity than RVFNIM and VIYKI. Interestingly, when compared to VIYKI, the RVFNIM data appears to be spread more evenly resulting in larger number of clusters compared to VIYKI despite showing smaller distance SD value. This suggests that VIYKI forms fibril classes that are very different but structurally and energetically similar within each class, whereas the RVFNIM fibrils are overall more similar across fibril classes compared to VIYKI fibrils. This may indicate that the energy landscape for the assembly reaction is flatter and more

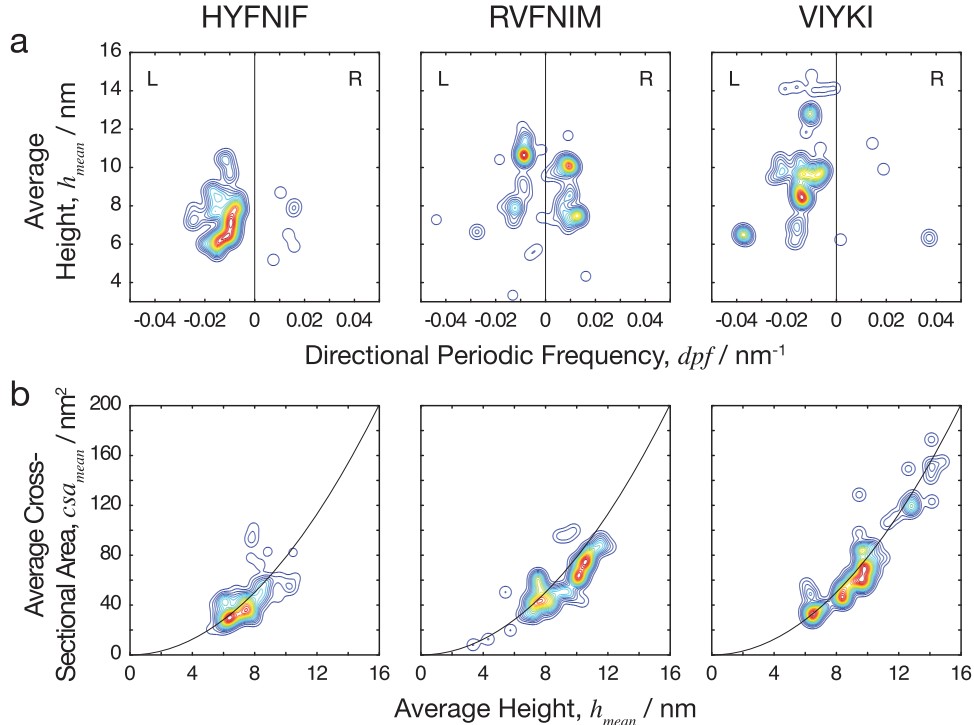

**Fig. 6 Comparing the heterogeneity of the polymorphic Waltz peptide assemblies. a** The average height of the fibrils plotted against the number of repeating units per nm, with negative and positive values to distinguish handedness (directional periodic frequency, *dpf*). **b** The average cross-sectional area of the fibrils plotted against the average height. The data is represented as a smoothed 2D histogram and visualised as a contour map, where the colouring represents the density of the data-points. The HYFNIF peptide assembly reaction favours one region, whereas the RVFNIM and VIYKI data was distributed across multiple regions. HYFNIF and VIYKI were also predominantly left-handed whereas the RVFNIM fibrils were almost evenly split between left and right-handed. See the Supplementary Fig. 4 for visualisation of the data in other pairs of structural parameters.

rugged for RVFNIM so that the structural differences between possible RVFNIM fibril polymorphs are on a more continuous scale compared with VIYKI fibrils with distinct but distant polymorphs.

## Discussion

High-resolution topographical AFM images of amyloid fibril samples assembled from three amyloidogenic peptide sequences using a gentle, force-distance curve-based AFM approach allowed for the detailed quantitative structural identification of polymorphs at a single fibril level. Individual 3D reconstructed fibril models were made for every traced fibril in the data set which included ~90 fibrils from each peptide assembly and 266 fibrils in total. This demonstrates the utility of AFM as an imaging method for the structural analysis of amyloid fibril populations at the single fibril level, and is highly complementary to cryo-EM methodologies. Morphometry is the process of measuring differences in the structures of objects and is often performed to measure differences in anatomy such as physiological differences in the brain[48,49]. Here, we perform quantitative nano-morphometric measurements on individual fibrils in heterogeneous amyloid samples. Using our AFM data, we were able to make quantitative measurements of different structural parameters which allowed us to distinguish between different polymorphs of amyloid fibril structure. Indeed, the data shows that each fibril in the amyloid samples is unique. The advances reported here open up the future possibility of analysing and comparing the structures present in entire populations of amyloid fibril species on a single fibril level, thereby opening up the possibility of linking population level and single fibril level properties. This connection may be key in order to decipher the relationships between specific polymorphs or structural

properties and phenotypic behaviours or biological consequences of amyloid.

The overall heterogeneity of the samples made from each different Waltz peptide precursor was enumerated as a single structural standard deviation value and quantitatively compared. The multi-parametric analysis results also provided objective classifications of polymorphs by measures of structural differences, and indicated the energy landscape associated with the filament assembly reaction of each peptide. Structural polymorphs that occur more frequently are expected to do so because they occupy a lower free energy state than other polymorphs[26,50,51]. Structural polymorphs which are different will have energy barriers separating the states. In the cases compared here, it was revealed that VIYKI assembly resulted in an overall more heterogeneous population than the other two peptides and that VIYKI and HYFNIF fibril assemblies are dominated by the formation of left-handed fibrils, whereas RVFNIM assemblies shows no overall preference to handedness as similar sized populations of both left-hand and right-hand twisted fibrils were formed. It has been thought that left-handed predominance in amyloid fibrils is a result of the natural curvature of the β-strands that make up the monomeric subunits which comes from the backbone of the polypeptide chains from L-amino acids[52]. However, it has been shown that show that fibrils of both left-handed and right-handed twist can be formed from the same sequence[53]. The results here supports the idea that preference for left-handedness is not absolute and the preference for handedness could be modulated by side-chain sequence[54,55]. In this case, the side-chains in RVFNIM assemblies may counteract the natural curvature of the β-strand due to the influence of specific side chains interactions in some polymorphic arrangements. Overall, the specific energetic explanation for the

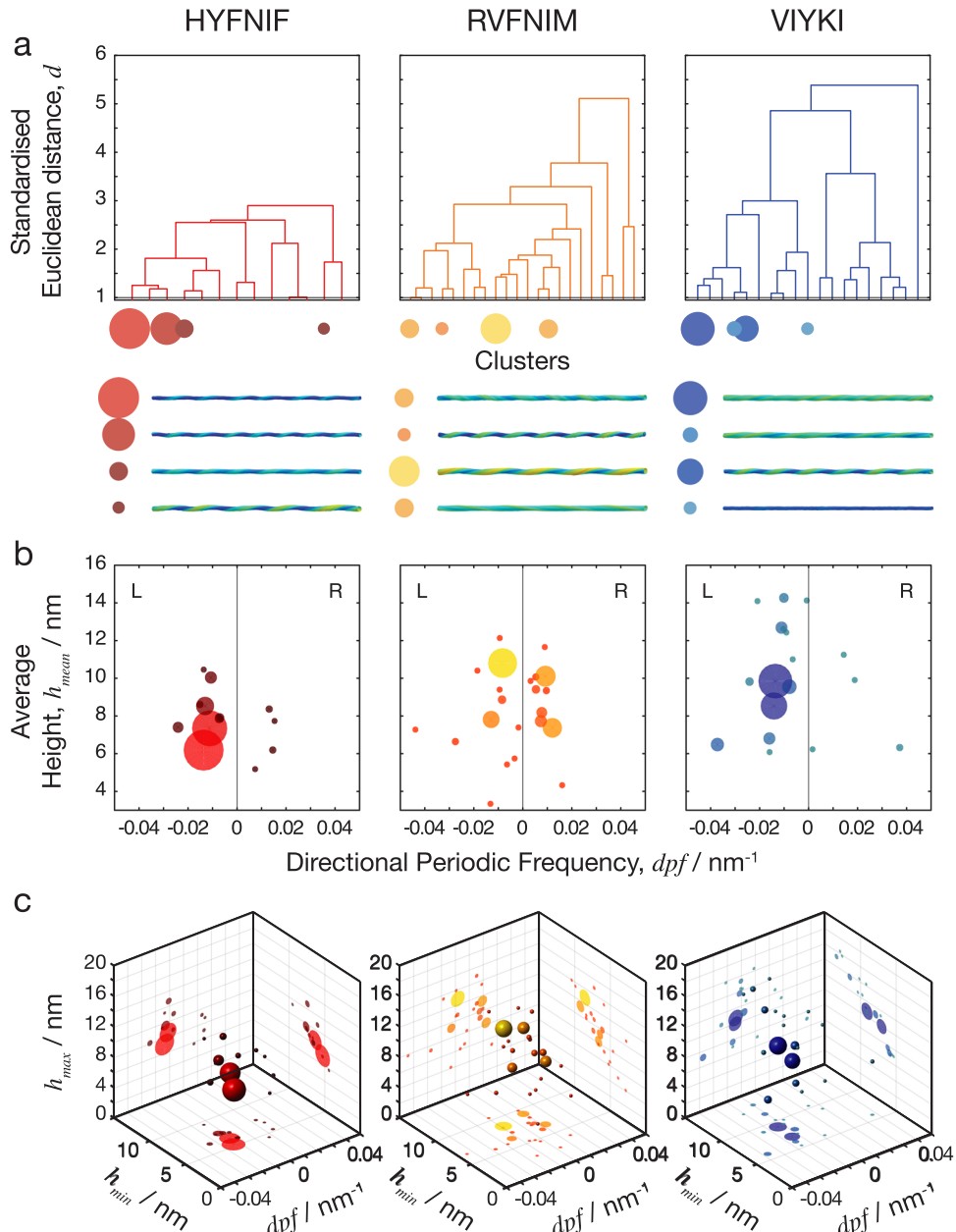

**Fig. 7 Natural separations in the clustered data define fibril polymorph classes.** The standardised Euclidean distance was measured in the 5-dimensional space between every possible pair of fibrils in each data set and the data was separated into clusters. **a** Dendrograms in which the x-axis represents the clusters that were generated, and the y-axis is the standardised Euclidean distance between each cluster. The four data clusters with largest numbers of fibril members are indicated as circles with size corresponding to cluster size, and typical fibril structural models for each of these clusters are shown below the dendrograms. In the dendrograms, the overall distance required to cluster each entire data set reflects the overall heterogeneity of the data. **b** Scatter plot of clusters shown as spheres for directional periodic frequency vs. average height where the data points are coloured dependent on the clustering at the cut-off level shown by the red line on the dendrograms above (1 standardised Euclidean distance). **c** 3D scatter plot of clusters, with directional periodic frequency vs. minimum height vs. maximum height with the same colouring of the clusters and projections of the various 2D plots projected onto the back walls of the plot. At this cut-off level, there were 13 clusters in the HYFNIF data set, 22 clusters in RVFNIM and 19 clusters in VIYKI with some clusters containing proportionally more of the data than others, here visualised as sphere size. Further visualisations of the hierarchical clustering analysis are shown in the Supplementary Figs. 5 and 6.

difference in behaviour between the three peptide assemblies remains unknown. However, the data here demonstrate the considerable influence of amino acid sequence in filament assembly reactions and opens up the possibility of systematic sequence-structure analysis of cross-β assemblies. Furthermore, utilising an agglomerative hierarchical clustering approach, we were able to objectively assign individual fibrils to different classes of fibril polymorph. By analysing the dendrograms that allowed

visualisation of the similarities and dissimilarities between individual fibrils, it was possible to identify the natural divisions in the datasets and existence of clusters that existed in the data. This information was then used to classify individual fibrils to a cluster. The number of clearly defined clusters also corroborated the overall heterogeneity of the data sets. The distance between the fibril classes may also reflect the possible plasticity of the types of polymorph observed in each data set. For example, VIYKI

fibrils are likely belong to a predominant type of polymorph, which is very different to any other type of possible VIYKI fibril polymorph. As a consequence, such a population if presented in biological context may not be able to shift to a different polymorph easily under pressure from environmental changes. The most common types of HYFNIF fibril, however, are very similar and so HYFNIF fibril populations may be able to shift their morphology much more easily when conditions change. Thus, characterising amyloid polymorphism by AFM[46,47] and quantifying structural differences on a single fibril level presents a possible means of comparing fitness of individual amyloid or prion polymorphs under any given set of conditions.

Structural polymorphism could account for some of the biological activities associated with amyloid observed in vivo. Individual fibril polymorphs will have different physical properties. For example, thicker fibril polymorphs with a circular cross section will have a different second moment of inertia compared to thin polymorphs with a more oval shaped cross section, which could result in differential stability to fibril fragmentation. Thicker fibrils with large surface areas per length may provide more sites for secondary nucleation events than thinner fibrils with small surface areas. Thus, fibril polymorphs displaying different properties would likely have different biological activities, even formed from identical precursors and present within the same population. In this case, individual fibrils within a population may also have higher rates of fragmentation that could result in a greater likelihood for propagation[56,57] or cytotoxicity[34], or higher rates of secondary nucleation that could result in more cytotoxic active species[38]. Various proteins and peptides with different amino acid sequences are amyloidogenic[58]. Yet, despite all of them being capable of forming amyloid fibrils in vivo, they display different biological activities[59–61]. Changing the primary sequence of an amyloidogenic protein, e.g., by point mutations also alters the energy landscape for that protein, resulting in structural polymorphism that could be responsible for variations in their biological or pathological activities. For example, recent AFM analysis of pathology associated point mutations in alpha-synuclein reveal considerable changes in structural polymorphism due to the point mutations[62]. Therefore, the variation in heterogeneity caused by polymorphism, as well as individual polymorphs could account for the differences in biological activity between different amyloidogenic precursors.

Prion and prion-like amyloid fibrils are able to propagate between cells[63,64]. Segregation of specific fibril polymorphs could result in the cell-specific propagation of polymorphs[65] likely manifest as strains in mammals or variants in yeast. This could result in different biological activities in different cells infested with the same amyloid sample. Examples of this include the identification of numerous strains of Tau, each of which causes a different pathology, in different brain regions and propagates at different rates[66] and the identification of two α-synuclein strains which show the same type of strain dependent phenomena[64]. This is of particular importance as some fibril polymorphs may react differently to potential inhibitors, which could result in strains that become 'resistant' to therapeutics over time. The strains phenomenon seen in prions and prion-like amyloid, in which structural characteristics presumably determine the biological activity of the fibrils has been demonstrated in human prions[67]. This behaviour has also been implicated in the Alzheimer's related Aβ peptides[68]. It is, therefore, important to investigate amyloid polymorphism in heterogeneous populations from a single fibril perspective, as this can account for differences in specific constructive or destructive biological activities associated with amyloid structures since each individual amyloid fibril particle has characteristics that may be vastly different to those of the population average.

## Methods

**Waltz peptide synthesis**. The Waltz peptides with the sequences HYFNIF, RVFNIM and VIYKI were purchased from JPT peptide technologies, or the Biomolecular analysis facility at the University of Kent. The peptides were synthesised with N-terminal acetylation and C-terminal amidation. Multistage solid phase synthesis using Fmoc protection chemistry generated a lyophilised powder with >95% purity as measured by HPLC.

**In vitro polymerisation**. To prepare amyloid fibril samples formed from the three Waltz peptides, 1 mg of the respective lyophilised powder was dissolved in 100 μl of filter sterilised milli-Q water to a final concentration of 10 mg/ml. The solution was incubated at room temperature for 1 week prior to imaging. This allowed for the 'maturation' of the fibrils from all three Waltz peptide samples to occur within a standardised amount of time.

**AFM sample preparation and imaging**. Each Waltz peptide assembly sample was diluted to 0.05 mg/ml in a dilute solution of HCl (pH 2.0, made up using filter sterilised milli-Q water). Immediately after dilution, 20 μl samples were deposited onto freshly cleaved mica surfaces (Agar scientific, F7013) and incubated for 10 min. Following incubation, the sample was washed with 1 ml of filter sterilised milli-Q water and then dried using a stream of nitrogen gas. Fibrils were imaged using a Multimode 8 AFM with a Nanoscope V (Bruker) controller operating under peak force tapping in the ScanAsyst mode using Bruker ScanAsyst probes (silicon nitride triangular tip with tip height = 2.5–2.8 μm, nominal tip radius = 2 nm, nominal spring constant 0.4 N/m, Bruker). The height channel images, each with a scan size of 6 × 6 μm and 2048 × 2048 pixels or 12 × 12 μm and 4096 × 4096 pixels were collected. A scan rate of 0.305 Hz was used with a noise threshold of 0.5 nm and the Z limit was reduced to 1.5 μm. The peak force set point was set automatically, typically to ~675 pN during image acquisition. Under the deposition and imaging conditions applied, no broken fibrils were observed. Care was also taken to minimise any deformation of the fibrils by the probe itself by manually ensuring the low peak-force were maintained during the scan and by using the slow scan speed. Nanoscope analysis software (Version 1.5, Bruker) were used to process the image data by flattening the height topology data to remove tilt and scanner bow.

**Structural data extraction**. Fibrils were traced[69,70] and digitally straightened[71] using an in-house application and the height profile for each fibril was extracted from the centre contour line of the straightened fibrils. The periodicity of the fibrils was then determined using fast-Fourier transform of the height profile of each fibril. Some fibrils do not display any clear periodicity and have a height profile that appears to show an erratic pattern so frequency of the highest peak in the frequency domain per nm was used. The final datasets consist of a varying multitude of images comprising ~90 individually traced fibrils per Waltz sequence. The same pixel density is maintained for all images within the dataset.

**3D modelling of fibril structures**. Straightened fibril traces were corrected for the tip-convolution effect using an algorithm based on geometric modelling of the tip-fibril contact points[72]. Briefly, the variation in the tip radius from their nominal value was first determined. Tip radii can be estimated by imaging standards such as gold nanoparticles with known dimensions[73]. It is also important to consider that the tip can become blunter with scanning action and the tip radius can widen over time. Therefore, the tip radius was estimated for each individual fibril on an image from the extent of convolution seen in data by assuming the twisted amyloid fibrils have ideal corkscrew symmetry with circular average cross-section of the fibril perpendicular to its axis of rotation. The algorithm then corrects for the lateral dilation of nano-structures resulting from the finite dimensions of the AFM probe, without the loss of structural information, by resampling of the fibril images to tip-sample contact points. This results in recovering subpixel resolution of lateral sampling. Each pixel value in the straightened fibril data is then corrected in their $x$, $y$ and $z$ coordinates. Filament helical symmetry was estimated by building 3D models with various symmetries from the data, back calculating a dilated AFM image and comparing the angle of the fibril twist pattern with that of the straightened fibril trace in the simulated images and in the 2D Fourier transform of the simulated images. Then for construction of the 3D models, the degree of twist per pixel along the y-axis was found by dividing 360° with the product of fibril periodicity and its symmetry number (e.g., 1, 2 or 3 etc.). The 3D models of the filaments were made using the structural information obtained on the top surface of the filaments. Assuming the filaments have helical symmetry then the bottom surface of the filament is structurally mirrored by the top part after an angular shift of the structural information along the fibril axis. This was accomplished using a moving-window approach, in which a window, centred at a pixel $n$, contained the pixels $n − x$ to $n + x$ where $x$ is the axial length covered by 180° twist. The central pixel $n$ is not rotated while neighbouring pixels on both sides along the y-axis are rotated by a rotation angle, which is the product of the twist angle and the distance from $n$ in pixels. Rotation angle values are negative in one direction from $n$ and positive in the other direction, with the specific direction depending on the handedness of the fibril, determined by manual inspection of the straightened fibril image and its 2D Fourier transform image.

**Nano-morphometric measurements on individual fibrils**. Fibril image datasets for RVFNIM, HYFNIF and VIKYKI were analysed with 92, 89 and 85 individually traced and characterised fibrils, respectively. Five morphometric parameters were extracted from the image data for each of the fibrils and the 3D models of the individual fibrils as shown in Fig. 5 The maximum, minimum and average heights were measured on the central ridge of each fibril. The periodic frequencies were obtained by Fourier-transform of the z-coordinates along the central ridge of each fibril. The average cross-sectional areas were obtained by averaging of the numerical polar integration along the fibril axis of the reconstructed fibril 3D models.

**Hierarchical clustering**. The standardised Euclidean distance $d(\mathbf{x}, \mathbf{y})$ in 5-dimensional space representing the five morphometric parameters average height, maximum height, minimum height, directional periodic frequency and average cross-sectional area (each data point consists of these five values, representing a single fibril segment) was calculated for every possible pair of data points (e.g., fibrils $\mathbf{x}$ and $\mathbf{y}$) using Eq. (1) where $\mathbf{V}$ is the 5-by-5 diagonal matrix whose $j$th diagonal element is the variance (standard deviation squared) for each of the five morphometric parameters. In Eq. (1), $\mathbf{x}$ and $\mathbf{y}$ are 1 by 5 row vectors representing each of the data points, with the elements in the vectors representing each of the 5 morphometric parameters for each individual fibril. Here, the diagonal elements of the $\mathbf{V}$ matrix contain the list of the variance for each parameter individually across all 3 data sets.

$$d(\mathbf{x}, \mathbf{y}) = \sqrt{(\mathbf{x} - \mathbf{y})\mathbf{V}^{-1}(\mathbf{x} - \mathbf{y})'} \qquad (1)$$

As expressed in Eq. (1), therefore, the Euclidian distances are standardised by dividing the distance in each of the morphometric parameter with their global standard deviation. Hence, the standardised Euclidean distance, $d$, between two fibril data-points is a unitless measure of the similarity (or dissimilarity) between the two. The distance standard deviation $SD$ of the morphometric data as a measure of the heterogeneity of the fibril populations was defined as the standard deviation of $d(\mathbf{x}, \bar{\mathbf{x}})$, which is the standardised Euclidean distance between each of the data points and the mean of all the data points, as shown in Eq. (2).

$$SD = \sqrt{\frac{\sum_{i=1}^{n}(d(\mathbf{x}_i, \bar{\mathbf{x}}))^2}{n-1}} \qquad (2)$$

In Eq. (2), $n$ is the total number of data points. Agglomerative hierarchical clustering was performed using the average linkage function shown in Eq. (3), which hierarchically links clusters using the average distance between all pairs of data points in any two clusters.

$$d(\mathbf{r}, \mathbf{s}) = \frac{1}{n_r n_s} \sum_{i=1}^{n_r} \sum_{j=1}^{n_s} d(\mathbf{x}_{ri}, \mathbf{x}_{sj}) \qquad (3)$$

In Eq. (3), $d(\mathbf{r}, \mathbf{s})$ is the average distance between cluster $\mathbf{r}$ with $n_r$ data points and cluster $\mathbf{s}$ with $n_s$ data points. Explained briefly, the shortest distance between any two data points within a data set is found. Those two data points are then considered to be a cluster, and the average coordinate of the two data points in the 5-dimensional space is used as the coordinate of the cluster. This is then repeated until all of the data is linked under one cluster.

## Data availability

All data generated or analysed during this study are included in this published article (and its supplementary information files).

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

## Acknowledgements

We thank the members of the Xue group, the LCS group, the CJS group and the Kent Fungal Group for helpful comments throughout the preparation of this manuscript. We also thank Ian Brown and Kevin Howland for technical support. This work was supported by funding from the University of Kent (B.J.F.B.), Biotechnology and Biological Sciences Research Council (BBSRC), UK grant BB/S003312/1, as well as Engineering and Physical Sciences Research Council (EPSRC), UK DTP grant (EP/R513246/1 for L.L.).

## Author contributions

L.D.A. and B.J.F.B. designed the research, conducted the experiments, and analysed the data. L.L. wrote the analytical software tools. C.J.S. and M.F.T. designed the research and analysed the data. L.C.S. designed the research, provided assembly reagents and methods, and analysed the data. W.F.X. designed the research, wrote the analytical software tools, analysed the data, and managed the research. The manuscript was written through contributions of all authors.

## Competing interests

The authors declare no competing interests.
