## [Peer Review File · Communications Chemistry]

Reviewers' comments:

Reviewer #1 (Remarks to the Author):

The Authors have done a remarkable work in investigating the polymorphism of amyloid fibers. They collected and analysed statistically relevant set of data on 3 peptides known to form amyloid like fibers.

Their analysis clearly demonstrates and quantifies the polymorphism. I think this is a manuscript that deserves publication.

However, I have some points that should be better clarified, namely the fact that the fibers are deposited on the substrate and therefore are deformed by adhering to the substrate. So how it is possible to recover what the authors name the "3D model"? Which are the steps that allow to "eliminate" the effect of the substrate? How the absorption might modify the recovered 3D model? This should be added to the present manuscript.

Then of course I am awaiting the more extended publication of the method of 3D reconstruction and deconvolution: the two and half pages (from line 404 to 450) are clearly not sufficient and I hope that the authors will soon publish a more detailed description of the methods used supported by data.

Another point is that the authors do not attempt to quantify in a mathematical model their results but I also agree that this goes probably beyond the scope of the present manuscript. In an article (Adamcik et al., Nature Nanotechnology, 5(2010) 423) these authors have looked into a basic physical model describing the pitch and persistence length of fibers. In the present manuscript no attempts are made to quantify by some physical model the difference between the fibers formed from the different peptides.

Reviewer #2 (Remarks to the Author):

The present manuscript is a detailed technical report on how AFM images on three series of amyloidogenic peptides can be analysed in detail to allow a 3D reconstruction of their polymorphic forms. It is a nice contribution, but in my view, the authors have not made sufficiently clear what is new in the story. For example, there is a whole section titled "Analysis of height, twist periodicity and cross-sectional area enables the quantification of heterogeneity due to fibril polymorphism" and in other parts of the text it is discussed how this analysis would allow 3D reconstruction of the different polymorphs. This, however, has already been done extensively in literature: see for example Adamcik et al. Nature nanotechnology 2010 for an original approach and Adamcik et al. Curr. Op. Coll. Interf. Sci. 2012 for a review on how AFM images allow reconstruction of polymorphism. So, the question: is there anything new here and if so, what is new? Authors need to do quite some effort to put their results in the context of what has already been done and explain differences, where these are present. Sentences such as "Thus, this work demonstrates a general utility to quantify polymorphism in amyloid fibril data sets by AFM on a single fibril level as a possible means to quantify and comparing fitness of individual amyloid..." are therefore way too general and very difficult to justify in view of the tremendous amount of work which was done before on the same topic.

Beside, some specific aspects need attention from the authors:

The discussion on handedness (It has been thought that left-handed predominance in amyloid

Fibrils... And the next 5 lines) sounds not very up to date. Handedness of final amyloids from the same peptide or protein sequence has been discussed extensively and it is now well understood that both handedness can occur from the same starting sequence. For a recent discussion see for example: R Jurado et al, JACS 2018.

It is not clear why the authors have made a selection of peptides such as HYFNIF, VIYKI and RVFNIM, which have little in common, also with respect to their primary sequence.

The AFM imaging mode is highly unclear as described. Authors mention Peak Force Tapping Mode. Do they mean the PeakForce-QNM (Peak Force Quantitative Nano Mechanics) mode from Bruker? If so, this method has already been applied to amyloids nearly 10 years ago (since 2011 more precisely), and the analysis should not be presented as new, rather, earlier works should be duly mentioned. Furthermore, why using PF-QNM and giving the output in height? This is at least my impression looking at Figure 2: I have to make my own extrapolation as the (height?) scale and the output (phase, height, Young modulus?) are not specified. Please revise this part accordingly.

There are several claims between the different polymorphs and their position in terms of energy landscape, but the way the discussion is made is very vague and qualitative and I see no real rationale to discriminate the energy level based on the polymorphism. This discussion could and should have been made sharper.

In summary, I think the work is well done and very sound on the scientific side in most of its parts; however the writing is often obscure, the state of the art is not really discussed comprehensively, some technical issues need to be addressed and I remain puzzled with respect to the novelty of these results and the described approach, at least following the style presently adopted in the manuscript.

Reviewer #3 (Remarks to the Author):

I have reviewed the paper "Quantification of amyloid fibril polymorphism by nano morphometry reveals the individuality of filament assembly" by Aubrey et al.

The paper represents a novel methodology for quantitative and accurate classification of amyloid fibril morphologies. The authors describe applications of AFM based imaging of fibrils formed from short model peptides and morphometric reconstitution of images and analysis thereof. Three short peptide fragments are used as model systems. Rather surprisingly these peptides show remarkable fibril polymorphism but this is a feature which is elegantly exploited by the authors. Image analysis of various structural features of the fibrils are clustered for grouping of features. In summary this is an excellent new method for very promising applications for other systems with importance for peptide design and disease associated fibrils. The method should be highly useful by the amyloid research field.

There are a few minor remarks:

1) The structural models used in Fig. 1 are based on previous data from some of the authors of the present paper. This is an elegant figure, but there is a risk that Fig. 1 is deceiving. The authors do not know that the remarkable fibril polymorphism observed is merely due to peptide packing. It is likely that also the conformation of the constituent peptides can be different in the various polymorphs. Hence it is advised that this possibility is clearly discussed in the paper.

2) It is advised that Supplementary Fig 2 replaces the subset of reconstructed 3D model fibrils shown in Fig 4 of the main paper. The reason being that the notable polymorphism observed should be highlighted rather than being eclipsed by a mere subset of structures shown in Fig. 4.

Point-to-point response to Reviewers comments:

Reviewer #1

The Authors have done a remarkable work in investigating the polymorphism of amyloid fibers. They collected and analysed statistically relevant set of data on 3 peptides known to form amyloid like fibers.

Their analysis clearly demonstrates and quantifies the polymorphism. I think this is a manuscript that deserves publication.

However, I have some points that should be better clarified, namely the fact that the fibers are deposited on the substrate and therefore are deformed by adhering to the substrate. So how it is possible to recover what the authors name the "3D model"? Which are the steps that allow to "eliminate" the effect of the substrate ? How the adsorption might modify the recovered 3D model ? This should be added to the present manuscript.

Reply: We agree that the requirement of adsorption of specimen molecules, in our case the amyloid fibrils, on a solid substrate support presents an important limitation of AFM. In this case as the reviewer eluded to, the surface substrate 1) obscures the bottom of the filaments from being observed and 2) may deform the filaments being probed. To address these potential concerns in the recovering of 3D models, we assumed a fibrillar model in which the filaments have helical screw symmetry (as independently confirmed for a number of amyloid fibrils using cryo-EM). Whilst the data only provide information about the top surface of the filament, we assume that the bottom surface of the filament structurally mirrors the top part after an angular shift along the fibril axis. In doing so, we also alleviate the concern that the bottom part may be deformed by adsorption to the surface support as the information used to recover the 3D model do not come from that part of the fibril. We have added this information to lines 14-18 on page 19 of the revised manuscript.

In addition, the adsorption process may also modify the fibrils as the reviewer said, resulting in damaged or broken filaments. This behaviour can be detected during AFM imaging, and such behaviours have been reported in other fibril forming systems (e.g. Figure 5 in Sun et al. J. Mol. Biol. (2008) 29, 1155-1167) However, we do not observe this effect for the filaments under the conditions used here. Finally, to further minimise any adverse effects resulting from the surface adsorption requirements of the AFM imaging experiments, we imaged the filaments using soft cantilevers under controlled low force to minimise any deformation from probing, as well as compared our images with negative-stain TEM images previously published on the same system as already mentioned in the manuscript. Therefore, we are confident that the recovered 3D models and our conclusions on the polymorphism by comparing our 3D models are as true to our fibril forming systems as possible. We have added this information in the revised manuscript at lines 4-7 on page 18 of the revised manuscript.

Then of course I am awaiting the more extended publication of the method of 3D reconstruction and deconvolution: the two and half pages (from line 404 to 450) are clearly not sufficient and I hope that the authors will soon publish a more detailed description of the methods used supported by data.

Reply: We are pleased to confirm that our manuscript addressing the technical details of our AFM reconstruction method is now published (*Biomol. Concepts* (2020) 11, 102-115, reference 71 in the revised manuscript) and can be accessed here: <https://doi.org/10.1515/bmc-2020-0009>.

Another point is that the authors do not attempt to quantify in a mathematical model their results but I also agree that this goes probably beyond the scope of the present manuscript. In an article (Adamcik et al., Nature Nanotechnology, 5(2010) 423) these authors have looked into a basic physical model describing the pitch and persistence length of fibers. In the present manuscript no attempts are made to quantify by some physical model the difference between the fibers formed from the different peptides.

Reply: We are aware of the article mentioned by the referee and other seminal articles by the same group of authors (including references 26 and 41 in the reference list of our original manuscript). We have now also included a reference to the publication mentioned in the revised manuscript (reference 47 on pages 7 and 15 of the revised manuscript). We agree that developing a mathematical/physical model to explain the polymorphic behaviours of amyloid is an important goal. However this is beyond the scope of this manuscript because such an endeavour would require quantification of polymorphism for a large number of different amyloid systems of varied sequences and amino acid chain lengths. Indeed, we propose that the methods we presented in this manuscript can be adopted by other laboratories with the expectation that a large dataset of amyloid polymorphism quantifications will soon emerge from our research community.

Reviewer #2

The present manuscript is a detailed technical report on how AFM images on three series of amyloidogenic peptides can be analysed in detail to allow a 3D reconstruction of their polymorphic forms. It is a nice contribution, but in my view, the authors have not made sufficiently clear what is new in the story. For example, there is a whole section titled “Analysis of height, twist periodicity and cross-sectional area enables the quantification of heterogeneity due to fibril polymorphism” and in other parts of the text it is discussed how this analysis would allow 3D reconstruction of the different polymorphs. This, however, has already been done extensively in literature: see for example Adamcik et al. Nature nanotechnology 2010 for an original approach and Adamcik et al. Curr. Op. Coll. Interf. Sci. 2012 for a review on how AFM images allow reconstruction of polymorphism. So, the question: is there anything new here and if so, what is new? Authors need to do quite some effort to put their results in the context of what has already been done and explain differences, where these are present. Sentences such as “Thus, this work demonstrates a general utility to quantify polymorphism in amyloid fibril data sets by AFM on a single fibril level as a possible means to quantify and comparing fitness of individual amyloid...” are therefore way too general and very difficult to justify in view of the tremendous amount of work which was done before on the same topic.

Reply: We thank this reviewer for their praise of the technical details and soundness in our work as well as their constructive criticism on how our wordings did not fully convey the novelty of the work reported in this manuscript.

As the reviewer pointed out, fibril polymorphisms has been observed using AFM imaging previously. “Analysis of height, twist periodicity...” that forms a part of our work, has also been reported in the literature by others. For example, in a very recent article Ruggeri et al., 2020 used analyses of the height and twist periodicity of alpha synuclein amyloid fibrils to yield important information on the different classes of alpha synuclein structural polymorphs in various pathogenic mutants (Ruggeri et al, ACS Nano 2020 March, added as reference 62 in the last line of page 15 in the revised manuscript). As already explained in our response to reviewer 1, we are also aware of the articles suggested by this reviewer as well as the seminal work done by the same group of researchers (references 26 and 41 in the original manuscript).

As suggested by the reviewer, we have now revised the sentence on page 14 of the original manuscript, taking into account all of the additional suggested references in the revised version of our manuscript, as follows (line 3-5 on page 15 in the revised manuscript): “Thus, characterising amyloid polymorphism by AFM [added references 46,47] and quantifying structural differences on a single fibril level presents a possible means of comparing fitness of individual amyloid...”

In terms of the novelty of the work reported here, while the articles mentioned by the reviewer have inspired the work we report here, we have in this manuscript made a step change in terms of how AFM can be used to analyse individual fibril particles, beyond what has been previously reported in the literature to the best of our knowledge. Firstly, we did not carry out “reconstruction” of our “polymorphic forms” or “reconstruction of polymorphism” as the reviewer said, but we 3D-reconstructed individual filaments based directly on the coordinates obtained from the AFM images. Therefore, our structural models are not conceptual models of the polymorphs as previously reported by the references mentioned by the referee, but reconstructions of the actual individual filaments based on the coordinates obtained from the AFM images. Secondly, we have devised a multivariate approach to quantify and classify the structural variation of amyloid fibrils within a population, enabled by the structural reconstruction of individual filaments. This type of approach has resulted in objective classifications of polymorphs by measures of structural differences. In other words, instead of assuming a number of distinct polymorphic classes and attempt to conceptually model the shapes of the assumed classes, we reconstructed individual filament structures directly from the data and let the structural differences inform an objective classification. We have now revised the relevant sections of the result and the discussion sections to highlight these two key innovations in our revised manuscript (page 7, line 6 from the bottom to line 5 page 8, and page 13, lines 17-22).

Beside, some specific aspects need attention from the authors:

The discussion on handedness (It has been thought that left-handed predominance in amyloid Fibrils.... And the next 5 lines) sounds not very up to date. Handedness of final amyloids from the same peptide or protein sequence has been discussed extensively and it is now well understood that both handedness can occur from the same starting sequence. For a recent discussion see for example: R Jurado et al, JACS 2018.

Reply: We thank the reviewer for highlighting this reference and have now revised our discussion on handedness and added the suggested reference accordingly (at line 6-7 on page 14 and reference 53 in the revised manuscript). Our work agrees with the conclusions of the suggested paper in that we also conclude that both left-handed and right-handed twists are possible for amyloid fibrils and the primary sequence influences the preference for handedness. In addition, our analysis suggests a sequence dependence for the relative abundance of the different handedness.

It is not clear why the authors have made a selection of peptides such as HYFNIF, VIYKI and RVFNIM, which have little in common, also with respect to their primary sequence.

Reply: The three sequences were specifically chosen for investigating the polymorphism as a fundamental features of amyloid forming proteins because they are very different but yet all three were identified as amyloid forming sequences by the WALTZ algorithm. These sequences have also shown to be polymorphic by negative-stain TEM and their amyloid forming cores have been previously characterised by X-ray fibre diffraction. We have now revised the manuscript to clarify these points (line 2 from the bottom on page 5 to line 3 on page 6 in the revised manuscript).

The AFM imaging mode is highly unclear as described. Authors mention Peak Force Tapping Mode. Do they mean the PeakForce-QNM (Peak Force Quantitative Nano Mechanics) mode from Bruker? If so, this method has already been applied to amyloids nearly 10 years ago (since 2011 more precisely), and the analysis should not be presented as new, rather, earlier works should be duly mentioned. Furthermore, why using PF-QNM and giving the output in height? This is at least my impression looking at Figure 2: I have to make my own extrapolation as the (height?) scale and the output (phase, height, Young modulus?) are not specified. Please revise this part accordingly.

Reply: PeakForce tapping refers to the force-distance curve-based imaging technology marketed by Bruker on their instruments. Both the ScanAsyst and the PeakForce-QNM modes are based on the PeakForce tapping technology. In Figure 2, height images are shown as already indicated by the figure legend in the original manuscript (page 27-28, “*The colour scale represents the height range from 0 to 12.5 nm and the scale bar represents 1 μ m in all images*”). The images were collected using the ScanAsyst imaging mode and not the PeakForce-QNM mode (i.e. we did not collect the Young’s modulus/dissipation/deformation etc. channels). The PeakForce-QNM experiments to gain reliable information on the nano-mechanical behaviours of the fibrils requires high enough contact force to generate a certain amount of deformation (typically \sim nm) and require a sufficiently stiff probe, whereas in our gentle imaging experiments we wanted to minimise contact force to minimise any deformation of the fibrils by the probe. Therefore, the ScanAsyst mode was used together with soft imaging probes with 0.4N/m nominal spring constant (Materials and methods in the original manuscript). We have now revised the description of PeakForce Tapping to include the name ScanAsyst and changed mentions of “force curve based” to “force-distance curve-based” throughout the manuscript to clarify this.

There are several claims between the different polymorphs and their position in terms of energy landscape, but the way the discussion is made is very vague and qualitative and I see no real rationale to discriminate the energy level based on the polymorphism. This discussion could and should have been made sharper.

Reply: While we have been very careful in suggesting the contour diagrams are not energy landscapes but “provide an indication of the energy landscape associated with the filament assembly reaction” (page 13 of the original manuscript), we thank the referee for pointing out that this may sound vague. The link between the populations measured and visualised in the contour diagrams and the free energy landscape of assembly comes from the fact that the probability that a system will be in a certain (assembly) state is a function of that state’s free energy at a given temperature as given by the Boltzmann distribution. We have now revised relevant sentences (page 9 and 13 in the original manuscript) to clarify this point (lines 2-4 from the bottom of page 9 and lines 4 and 5 from the bottom of page 13 in the revised manuscript).

As per the discussions in the original manuscript, different polymorphs have different physical features such as fragmentation rate etc., which can impact the biological and pathological properties of the fibrils. For example, pathogenic mutations in alpha-synuclein resulting in different structural polymorphs, as recently reported by the recent article by Ruggeri et al ACS Nano 2020. We have now revised these sections and added this additional reference to further clarify the biological impact of assembly landscapes (lines 1-6 from the bottom on page 15 in the revised manuscript)

In summary, I think the work is well done and very sound on the scientific side in most of its parts; however the writing is often obscure, the state of the art is not really discussed

comprehensively, some technical issues need to be addressed and I remain puzzled with respect to the novelty of these results and the described approach, at least following the style presently adopted in the manuscript.

Reply: We thank the reviewer again for the positive feedback as well as constructive criticisms. We hope the reviewer agree that our revised manuscript has sharpened the presentation of the novelty of our work in terms of the novel individual filament 3D structural reconstructions and the objective multivariate structural analysis.

Reviewer #3

I have reviewed the paper "Quantification of amyloid fibril polymorphism by nano morphometry reveals the individuality of filament assembly" by Aubrey et al. The paper represents a novel methodology for quantitative and accurate classification of amyloid fibril morphologies. The authors describe applications of AFM based imaging of fibrils formed from short model peptides and morphometric reconstitution of images and analysis thereof. Three short peptide fragments are used as model systems. Rather surprisingly these peptides show remarkable fibril polymorphism but this is a feature which is elegantly exploited by the authors. Image analysis of various structural features of the fibrils are clustered for grouping of features. In summary this is an excellent new method for very promising applications for other systems with importance for peptide design and disease associated fibrils. The method should be highly useful by the amyloid research field.

There are a few minor remarks:

1) The structural models used in Fig. 1 are based on previous data from some of the authors of the present paper. This is an elegant figure, but there is a risk that Fig. 1 is deceiving. The authors do not know that the remarkable fibril polymorphism observed is merely due to peptide packing. It is likely that also the conformation of the constituent peptides can be different in the various polymorphs. Hence it is advised that this possibility is clearly discussed in the paper.

Reply: We thank the referee for their positive and supportive comments. We agree on this point and have revised the figure legend and main text (line 5 from the bottom on page 5 in the revised manuscript) accordingly to clarify this point.

2) It is advised that Supplementary Fig 2 replaces the subset of reconstructed 3D model fibrils shown in Fig 4 of the main paper. The reason being that the notable polymorphism observed should be highlighted rather than being eclipsed by a mere subset of structures shown in Fig. 4.

Reply: We agree with the reviewer and we have swapped Figure 4 and Supplementary Figure 2 in the revised manuscript.

REVIEWERS' COMMENTS:

Reviewer #1 (Remarks to the Author):

Editor's note: this reviewer provided no comments to the authors.

Reviewer #3 (Remarks to the Author):

This paper demonstrates an excellent methodology for imaging of individual fibrils and would have important applications for other systems. The approach will be extremely useful for the amyloid research field. Especially this approach used in the context of complementary methods we will be closer to understanding fibril polymorphism and disease mechanisms.

My initial concerns of the original submission have been met and I consider their overall the response to the referees comments to be well justified. Appropriate credit to previous work has been included.